# A Machine Learning-Based Model for Epidemic Forecasting and Faster Drug Discovery

Konstantinos D. Stergiou, Georgios M. Minopoulos, Vasileios A. Memos, Christos L. Stergiou, Maria P. Koidou and Konstantinos E. Psannis *

Department of Applied Informatics, University of Macedonia, 54636 Thessaloniki, Greece
* Correspondence: kpsannis@uom.edu.gr; Tel.: +30-2310-891-737

**Abstract:** Today, healthcare system models should have high accuracy and sensitivity so that patients do not have a misdiagnosis. For this reason, sufficient knowledge of the area is required, with the medical staff being able to validate the correctness of their decisions. Therefore, artificial intelligence (AI) in combination with other emerging technologies could provide many benefits in the medical sector. In this paper, we demonstrate the combination of Internet of Things (IoT) and cloud computing (CC) with AI-related techniques such as artificial intelligence (AI), machine learning (ML), deep learning (DL), and neural networks (NN) in order to provide a useful approach for scientists and doctors. Our proposed model makes use of these immersive technologies so as to provide epidemic forecasting and help accelerate drug and antibiotic discovery.

**Keywords:** artificial intelligence; deep learning; drug discovery; epidemic diseases; forecasting; machine learning; neural networks

## 1. Introduction

In recent years and especially nowadays, there has been an outbreak of viruses that have spread around the world, with a high mortality rate, such as the novel Coronavirus. To develop new treatments against widely spread viruses [1], the use of machine learning (ML) may facilitate an effective virtual examination of vaccines aimed at eliminating the virus spread. In various aspects of pharmaceutical exploration ML approaches have already been used with success, such as predicting drug interactions, predicting drug combinations, and predicting drug toxicity [2].

New scoring functions have been introduced in the field of virtual sorting based on recent models of ML regression and have been shown to exceed a wide range of functions. The results based on ML show a substantial improvement in the classic scoring functions in both ranking power (relative ranking prediction) and scoring power (commitment relevance prediction). Moreover [3], ML scoring functions that use the random forest algorithm or regression reinforcement trees were more often correlated with better efficiency. By accumulating data in shared databases, it is feasible to design score functions for the most effective prediction [4].

Epidemic forecasting and drug discovery constitute a complex and multi-factorial dependent fact [5], which takes a long time to be executed. ML provides methods and tools that can improve epidemic forecasting and drug discovery by evaluating big and high-quality input data. Occasions to utilize ML arise in every phase of drug discovery. Therefore, an ML-based model can offer accurate predictions and insights.

Many studies have been conducted in the area of drug and antibiotic discovery that promise an optimized method for finding and accelerating this procedure thanks to ML approaches [6]. A promising DL method for antibiotic discovery, which may be very useful in medicine, is presented in [7], where deep neural networks (DNNs) are able to predict molecules with antibacterial activity.

Moreover, virtual screening (VS) is regarded to be a sophisticated and useful method in the drug development process, while ML methods in combination with VS can be used for drug leads. In addition [8], ML methods can be used for VS and applications for drug discovery against Alzheimer's disease. It is a fact that ML for VS generally includes the reconstruction of a filtered training set of chemical compounds, which consists of known actives and inactives. After the model training and validation [9], if it is accurate enough, it can be applied to previously unseen databases to screen for new chemical compounds with the preferable drug activity.

A combination of ML and digital microfluidics was used for drug discovery and development, increasing throughput and reliability by computationally designing algorithms. Furthermore, by using appropriate deep learning algorithms, the time and economic costs in traditional drug research can be eliminated, which is proposed in more.

Finally, the unfortunate situation of COVID-19 and its tremendous contagiousness has forced many researchers to study and remedy it through novel technologies. ML techniques can be used as a "weapon" to understand and fight this virus. In [10,11], ML algorithms seem to be efficient in the diagnosis of COVID-19 and prediction of mortality risk and severity. The results indicate that ML qualifies a reasonable level of precision in the diagnostic and prognostic features of COVID-19 mortality. Another issue related to the spread of the disease is the ability to predict future forecasting. In [12], ML modelling shows the competence to predict the number of people who are affected by COVID-19 as a potential threat to human beings. Thus, by analyzing data sets containing real data from the previous days, ML algorithms are able to make forecasts about future days. Long COVID symptoms can be identified as well through ML. The examination of patients' symptoms used in order to develop ML models to identify potential patients with long COVID was described in [13].

The rest of this paper is structured as follows: Section 2 presents our proposed approach to evaluating the most-known cutting-edge methods; Section 3 describes the results of our proposed system architecture; and finally [14], Section 4 concludes the paper and gives some potential future directions.

## 2. Methods

There are many challenges to applying ML in medicine, especially for faster epidemic forecasting and drug discovery. Thus, an effective scheme that involves cutting-edge technologies is imperative to achieve this goal. Our proposed model is the integration of six emerging technologies that can constitute a robust system that will have improved capabilities in the detection of dangerous and fatal viruses, such as SARS-CoV-2 [15]. These technologies are presented in Figure 1 and analyzed below.

The backend deploys a deployable unit of work configuration containing a Docker container and then runs the ML model for training. The Flow connector expects a stream at this point, so training cannot start until the Kafka topic has a stream. We use at least two Kafka topics to solve this problem [16]. One or more data subjects containing only the training and evaluation data streams are required for training and evaluation. Machine learning models implemented by control subjects are notified by control messages, indicating when and where the data streams can be used for training and assessment. The training data stream must be sorted for delivery according to the data formats available in ML. Output information parameters that define the location of the data stream output [16] when sending a control message to the control header are set in the ML destination of the data flow.

Configuration and subsequent ML models receive control messages they know exactly where to deserialize the data stream in Apache Kafka and how many data streams to use for training and evaluation. Machine learning currently supports RAW for serialization and Apache Avro is supported [17]. However, it supports newer data formats. In each case, the unbundling information is included in the control message such as the training and tag data schema in Avro format. We have developed libraries for both of these data

formats that make scheduling streams easier for users, as they handle aspects of Kafka-ML such as sending control messages during data streams. Without this library, users have to manually send data to Apache Kafka by performing deserialization techniques and send control messages as described earlier.

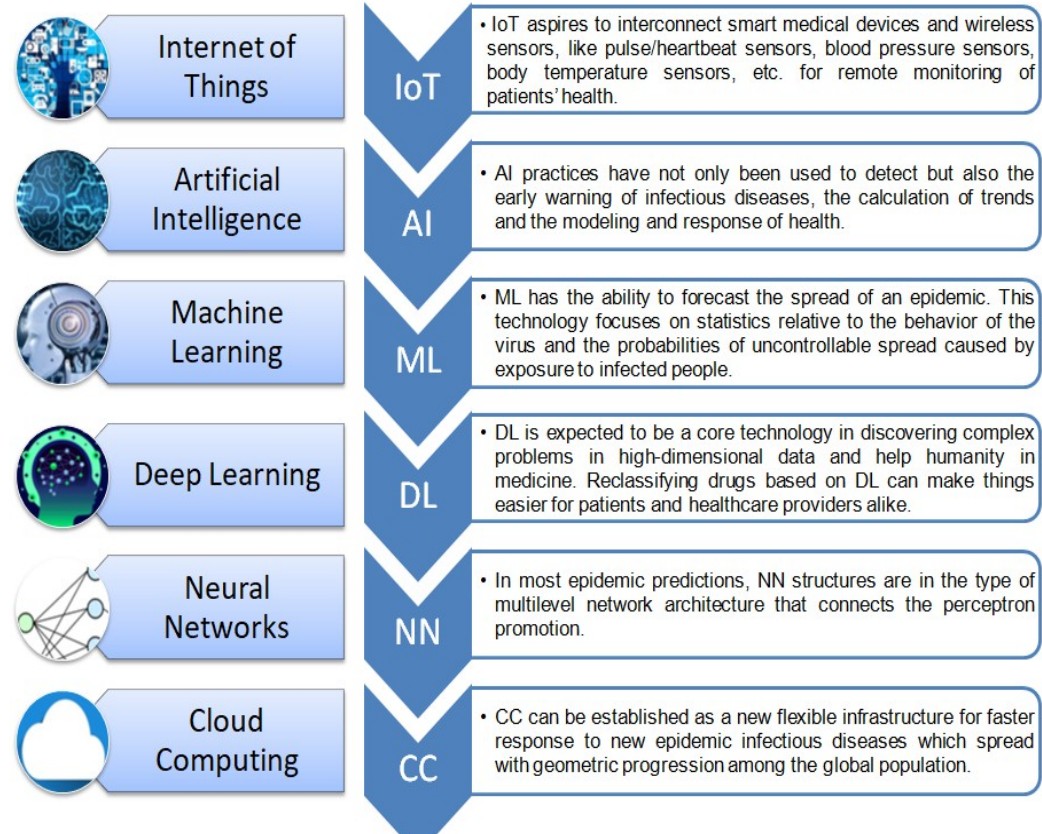

**Figure 1.** The emerging technologies used in our proposed system.

Algorithm 1 describes the process of training work that has been run by the ML back-end and shows a sequence diagram of the training process. It should be noted that some steps such as handling algorithm connection exceptions and decoding data streams are not included for convenience [18]. The ML model is downloaded from the back-end only if the training function is installed by the back-end and loads it as a ready-to-train TensorFlow model. It then creates a consumer, which waits for this process to receive control flow. This matches the deployment_id received by the back-end in the job. The control message determines how the data stream sent by the client will be deserialized for the batch size used for training and where the streaming data is available.

If the error is caused by something other than the data flow, the component will restart and reprocess the control message that was received earlier. After the data stream is received, the training and evaluation dataset is divided according to the validation rate given in the control message so that this parameter represents the number of evaluations in the form of the data stream. If msg.validation_rate is set to 0, only training and evaluation are executed, i.e., the entire data stream goes to the training stream variable. So, if only training is enabled, msg.validation_rate will be set to 0 when the control message is sent. The model is then alternately trained and evaluated. Note that ML does not allow continuous learning, which means that the training is done in one go and does not constantly accumulate knowledge from past data streams. Therefore, if a small stream of data is received, the ML model will only use these data for training [6]. The availability of data streams for training tasks is controlled through control messages so that ML users can appropriately check

when the aggregated data streams are ready for training. The data flow used for training must be at least larger than the batch size specified in the sent control message. Finally, the paper presents the trained model along with the training and assessment metrics in the background. This operation is performed in each function of each ML model in the configured configuration.

---

**Algorithm 1** Training algorithm.

---

```
while not trained do
    msg ← readControlStreams();
    if deployment_id == msg.deployment_id then
        decoder = getDecoder(msg.input_format,
        msg.input_config);
        data_stream ← readStream(msg.topic, decoder,
        msg.batch);
        if msg.validatition_rate > 0 then
            training_stream ← take(data_stream, (1 -
            msg.validation_rate));
            evaluation_stream ← split(data_stream,
            msg.validation_rate);
        else
            training_stream ← data_stream;
        end
        training_res ← trainModel(model, training_kwargs,
        training_stream);
        if msg.validatition_rate > 0 then
            evaluation_res ← evaluateModel(model,
            evaluation_kwargs, evaluation_stream);
        end
        uploadTrainedModelAndMetrics(model, training_res,
        evaluation_res);
        trained ← True;
    end
end
```

---

- Internet of Things (IoT)

Recent developments in information and communication technologies have allowed the interconnection of a growing number of devices on the internet. A cutting-edge technology is the Internet of Things (IoT) that aspires to interconnect physical objects such as smart devices to the internet, generating a global network where humans can interact with the smart devices [18]. In medicine, such devices include medical devices and wireless sensors, such as pulse/heartbeat sensors, blood pressure sensors, noninvasive oxygen saturation ($SpO_2$) sensors, electrocardiography (ECG) sensors, glucometer sensors, body temperature sensors, etc. for remote monitoring of patients' health.

As it is obvious, these IoT-based smart devices can have artificial intelligence, since they are able to make decisions based on their measurements to provide important information about the health level of the patients, while they can also access e-health data gathered by other smart devices [19]. All of these e-health data can be transmitted to medical laboratories after patients' consent, thanks to IoT technology and wireless networking, in order to be analyzed and used for drawing vital conclusions that can lead to faster drug discovery against specific epidemic diseases.

In addition, algorithms and tree protocols can be collected to track the path of an IoT network, and thus map the spread of the virus. Compression of the computer virus and biological disease in a weighted network has attracted the attention of researchers in various fields. Specifically, the global medical community studies the deceleration ways of the epidemic spread in weighted networks, such as the IoT.

- Artificial Intelligence (AI)

Artificial intelligence (AI) is the integration of human intelligence into machines. Machines work according to certain rules defined by appropriate algorithms to solve problems. Because of the "intelligent" behavior of such machines, the technology is called artificial intelligence. Thanks to artificial intelligence, machines are able to move and manipulate objects, recognize when someone raises their hand, and often solve multiple problems.

Over the past ten years, artificial intelligence technology has entered an astonishing period of rapid development and widespread application due to the significant improvements in raw computing power. The methods used in conventional AI study topics, such as computer vision, natural language processing, speech recognition, and robotics, have found many innovative applications in a number of real-world applications. Public health supervision is an area that has been advanced considerably by the latest AI developments. AI practices have been used not only for the detection but also early warning of infectious diseases, the calculation of trends and the modeling and response of health. Such large-scale public health supervision and response work presents distinct technical challenges. Conventional health supervision depends heavily on statistics. In recent years, there has been a remarkable development of methodologies with the possibilities of AI that complement statistical approaches [20].

AI unlocks the access to use it with an array of innovative or unexplored sources for public supervision objectives, particularly those not initially or deliberately intended to counter epidemiological subjects. For example, with the speedy improvement of the internet and IoT applications, pervasive social and device sensing potentials are becoming realistic, appearing with noteworthy surveillance capabilities. A diversity of open data, external to old-fashioned health supervision schemes, can be prolifically utilized to enrich surveillance capabilities. Health supervision is concerned with the early and effective evaluation of the threat of an epidemic, spotting irregular behavior in the spatiotemporal condition of the epidemic, so as to make available timely notifications and estimates concerning the trend of the epidemic.

Furthermore, AI delivers modeling frameworks for simulating data and multiple scenarios of disease spread and health reactions. The evolution of epidemics over time and in humans has a high degree of difficulty and uncertainty. The spread of the infection is non-linear in nature, with vague features. Furthermore, AI delivers framework models for simulating complex requirements and scenarios for the spread of diseases in public health responses. The progress of epidemics presents to a great extent the difficulty and uncertainty. The spread of infection is non-linear in nature, with vague characteristics and predictability. Therefore, the models are based on aggregate statistics and linear interactions. An AI subdomain, multi-factor systems, provides modeling frameworks that make available the analysis of epidemic development under different conditions [20].

- Machine Learning (ML)

Machine learning is a field of exploration devoted to understanding and developing styles of "learning", a system of using data to ameliorate the performance of a range of tasks. It is considered part of artificial intelligence. Machine learning algorithms make models grounded on sample data (called training data) to make prognostications or opinions without being explicitly programmed. Machine learning algorithms are used in various functions like those used in drug discovery, speech recognition, and computer vision. In cases that are sensitive or intractable, traditional algorithms are used to perform the necessary tasks. A subset of machine learning is closely related to computer statistics, which focuses on using computers to make prognostications, but not all machine learning is statistical literacy. The study of fine optimization provides proposition, and operation areas for machine learning. Data mining is an affiliated field of study that focuses on exploratory data analysis through unsupervised learning. Some executions of machine learning use data and neural networks in ways that mimic the way biological brains work. In its cross-business operations, machine learning is also known as predictive analytics.

In computer science, machine learning (ML) is considered as a subfield of AI. An ML algorithm defines any computational process where the output from previous decisions or actions is exploited to enhance the final decisions or predictions. Today, ML approaches are widely used in research, since they offer an automation in multi-scale biological data analysis [21].

ML can be separated into three categories: supervised learning, unsupervised learning, and sequential learning. The goal in supervised learning is to label new observations. The unsupervised learning identifies underlying relationships or patterns. In sequential learning, the algorithms repeatedly utilize external observations to attain the optimal decision about the environment in which they interrelate [22].

ML algorithms, by analyzing the environmental data in regions where population is affected, have the ability to forecast the spread of an epidemic. This technology focuses on statistics relative to the behavior of the virus and the probabilities of uncontrollable spread caused by exposure to infected people. The prediction is feasible by considering the virus spread among the contagious people within the specific area [23]. Thus, high-fidelity prediction can be applied by using ML techniques to predict a virus' dissemination. AI focuses on developing mathematical models to analyze this epidemic situation using common data nationally. Additionally, with the help of ML, it will be possible to separate volumes that belong to a high-risk group and to people who do not belong. As a result, a better technique in preventing and protecting people is applied. With the use of ML and DL models, it will be possible to understand the daily exposure behavior, as well as predict the future availability of viruses in all nations, using the information in real-time.

- Deep Learning (DL)

Deep learning (DP) is a method in which feature levels are not designed by humans to be able to be learned from data using a general learning process. In-depth learning makes significant progress in solving problems that have perplexed the AI community for several years. DL is expected to be a core technology in discovering complex problems in high-dimensional data [24] and help humanity in many areas of science, such as medicine, as well.

Recently developed artificial intelligence methods, such as deep learning and related modeling studies, provide new solutions for the efficacy and safety assessment of drug candidates based on big data modeling and analysis. Links between drug candidates and their possible adverse drug effects or new uses are often difficult to predict because the exact principles driving them are largely undefined, complex, or scattered and hidden in bodies of knowledge. The discovery of multi-domain modules and pharmacologically useful biomolecular subnetworks and pathways from database collections by independently mining multiple datasets is a hot topic of research.

Developments in DL algorithms create effective data models, and the proven accomplishments of these methods in many public tenders have contributed to the huge growth of DL applications within pharmaceutical enterprises in the last couple of years. DL uses sophisticated algorithms to generate systems that can detect features from a vast amount of untrained or labeled training data [25].

However, other than saving lives, reclassifying drugs based on DL can make things easier for patients and healthcare providers alike. First, patients will not be denied necessary medications due to stricter regulations on those classified as Schedule 2+. This reduces pain and suffering caused by strict regulations and allows sick people to recover more quickly. Second, doctors will not have difficulty prescribing necessary medications after learning exactly which ones they should avoid treating patients with—this reduces errors associated with guessing which medications are safe for patients to take. In addition, when rules on safe medication use are well-known by health professionals and patients alike, patient safety is significantly improved by highlighting any medication errors early on, before they cause any harm or fatalities.

Drug redefinition can be achieved effectively using in-depth learning methods. Reusing drugs based on DL is a cheaper, faster, and more effective approach and can minimize

clinical trial failures. There, the drug can enter directly into the advanced phase for testing without the initial tests and toxicity tests. Although DL-enhanced drug repositioning is currently in its infancy, this approach is a promising solution for the development of potentially therapeutic drugs.

- Neural Networks (NNs)

Neural networks (NNs) are complex methods developed to mimic the flow and learning of information in the human brain. These methods are extremely promising because they improve on any given scheme by combining multiple algorithms at the same time. In NNs, input characteristics are fed to an input level, and predictions are generated from an output level after a number of nonlinear transformations using hidden levels. This method most often uses the reverse propagation of errors, resulting in a gradual reduction of the price difference between received and expected. If there is only one output node, then you consider the network to be a NN [26].

Back-propagation is a popular technical approach, while it is one of the most extensively used learning algorithms in epidemic prediction. Given the fact that accurate forecasts help in the informed judgment for preventive intervention, in healthcare and epidemic control, this goal can simply be achieved using appropriate techniques and methodologies. As much as predictive accuracy is imperative, model selection methods and types are vital to predict accuracy.

The selection of the artificial neural network (ANN) methodology is essential for epidemic predictions with great accuracy. An ANN model is a mathematical evolutionary method of problem solving that provides solutions based on the concept of organization for transmitting signals to the human nervous system. It is a customizable processing model that receives arbitrary input and produces the result based on non-linear relationships between variables and parameters [27].

In the medical field, ANN has been applicable for clinical diagnosis, image evaluation and interpretation, and drug discovery. In epidemiology, ANN has effectively been used to examine the dynamics, growth, risk and control of infectious viruses and diseases. In epidemic forecasting, all layers and nodes of the network architecture are reconnected; it means that each neuron in one layer is connected to all neurons in the following layer.

The ANN architecture for predicting an epidemic depends on the problem, based on the variables and the parameters of the problem studied. In most epidemic predictions, NN structures are in the type of multilevel network architecture that connects the perceptron promotion. The interaction of causation, disease transmission, and control aspects in epidemic expansion tends to warn of the choice of network architecture. While the intent of epidemic control is to mitigate disease development at minimal cost [28].

- Cloud Computing (CC)

Cloud Computing (CC) is a novel architecture designed to provide resources and services stored somewhere on the internet, i.e., on cloud servers, or generally in the "cloud"; their users come from anywhere and from any device with internet access. Thus, CC provides many benefits for many branches of science.

This tech concept refers to the use of remote server and storage facilities via the internet. The new health care and education paradigm is emerging where people learn necessary skills using cloud-based platforms. Introduction cloud computing allows businesses to save money by using shared resources to run their operations. This tech approach has become popular among tech-savvy companies, as it reduces operational costs. Running a business this way is much more practical than leasing hardware from a provider or using an app-based solution. In addition, "bring your own device" (BYOD) models are becoming more common in schools as well. People can easily share cloud storage with colleagues or save documents directly to a remote server via apps such as Dropbox. Enterprises and individuals alike have found great benefits in adopting cloud technology recently.

In medicine, CC is regarded to be a recent and rapidly growing field of healthcare development. Thanks to its features [29], CC can be established as a new flexible infras-

tructure for faster response to new epidemic infectious diseases that spread with geometric progression among the global population. This is due to the fact that CC resources can be accessed in real-time simultaneously by multiple medical centers, and useful information (e.g., about a drug discovery) can be shared amongst them for faster response to epidemic diseases.

CC is often used to make calculations in several areas, such as in genomics, proteins, and molecular medicine. Nevertheless, other areas of CC application, such as health information systems, data exchange, image processing, or health management, are still under-represented [30].

The concept of health care is considered to involve all of the procedures that are relevant to the diagnosis, treatment, and prevention of human diseases or injuries, as well as clinical research and management of health care. CC has been proposed as a new business model for the exchange of biomedical information, which can benefit health services and describe opportunities and challenges. Therefore, it is feasible to identify the current situation and vital issues in CC healthcare research beyond the conventional area [31].

## 3. Results

With the advancement of AI, it would be feasible to discover new antibiotics with possible clinical applications. From the branch of medicine, it is known that a drug or antibiotic should pass four (4) steps in order to be launched in the market. These steps are depicted in Figure 2. Firstly, the granting of pharmaceutical compounds is necessary. This granting should be sufficient for the molecular goal defined by the medical community. Notably, the goal should be specific and take into account all parameters from the external environment (information input). With the required "know-how" of the scientific research group, the drug discovery phase can be achieved. However, it is important to conduct drug trials aimed at specific target groups and with specific clinical trials in order to evaluate and predict the action of the produced compound. Thus, the produced drug can be improved and approved for market circulation if it achieves optimal rates of effectiveness. Otherwise, it will be rejected and its distribution will be canceled. It is obvious that the abovementioned emerging technologies should be integrated and used properly in order to achieve the desired results.

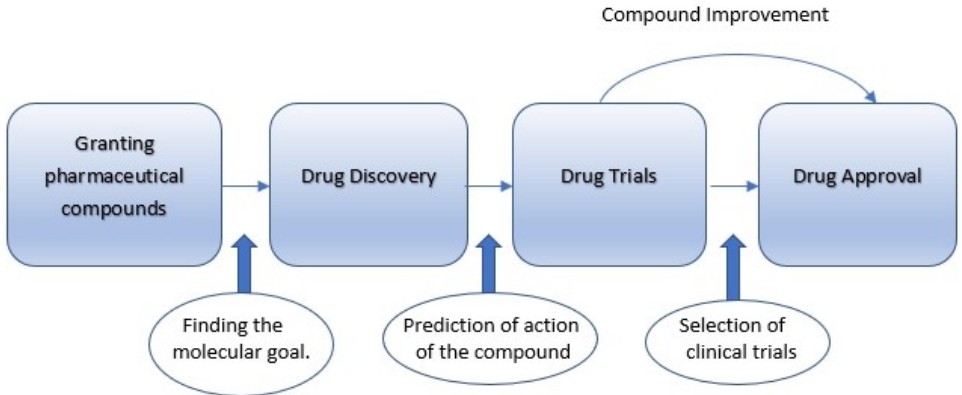

**Figure 2.** The steps for a drug to be launched in the market.

Bacteria have been around since the creation of the planet, so they have been on Earth for billions of years. This has led to their evolution and consequently their increased resistance to drugs and antibiotics. So, all research doctors are trying to develop new treatments to treat the mutated infections. Through research, they try to discover new antibiotics that will be more effective and focus on the specific bacteria without having any side effects. Advancements in technology and specifically in chemistry will help in the discovery of new antibiotics from candidates that can consist of thousands of chemical

compounds. In Figure 3, we can see how AI technology can suppress the growth of an infectious virus [32].

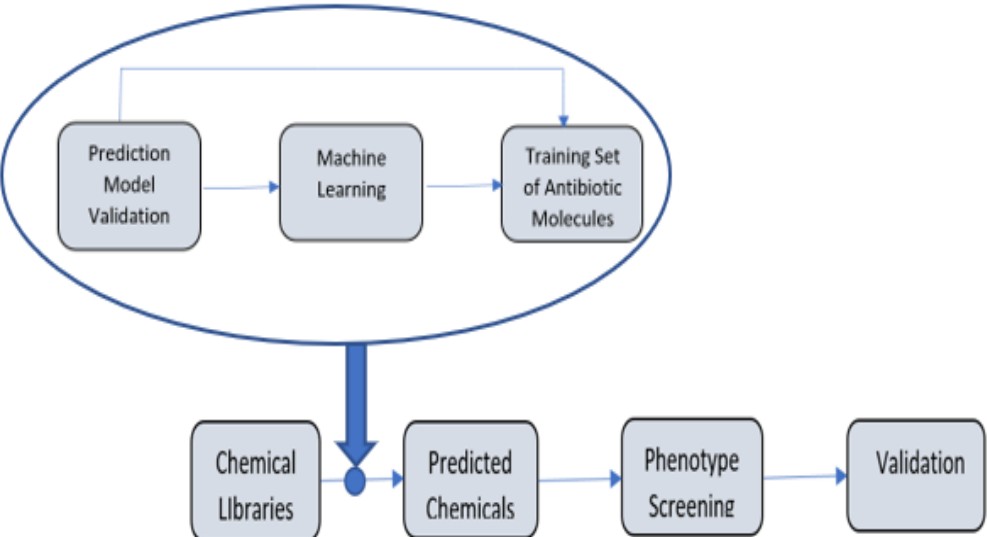

**Figure 3.** The proposed ML approach for accelerating drug and antibiotic discovery.

In supervised learning, the desired output in addition to the inputs is introduced into the neural network. Thus, each network is powered by input and output vectors. In the first stage, the values of the synaptic weights are randomly selected, and for each new input vector, they change until they converge on the real one with the desired network response to the desired level. This difference is defined as an error signal. The signals that reach the output level in case of unwanted convergence are fed backward. For the purpose of optimal network training, many vectors are introduced into the input layer. The learning process is completed when the weights do not show significant changes.

The inability of the network to solve a problem is addressed by the intervention of the trainer, who reshapes the structure of the network so as to achieve a proper way of operating it. With training, the weights will be stable, during the phase of which the neural network is able to deal with its environment on its own without the interference of an instructor. To evaluate the performance of a network that follows this form of learning, the mean square error, or the sum of the squares of the errors, is selected. The graphical meaning of this function is summarized in an association between an error surface and system performance. Every function of this system is located in this correlation with a point. The closer to a local minimum the point moves, the better it approaches the optimal performance.

Therefore, during supervised learning, the instructor defines and has knowledge of the input vectors entering the network, while he also sets the target output. A learning system with an instructor under appropriate conditions, such as a significant number of repetitions and training time, in combination with a well-designed algorithm, is capable of achieving the desired response.

The model is determined by molecular characteristics, optimizing the hyperparameters and the set, which lead to the final DNN stage through the continuous training of the model. Compared to the conventional method, the AI approach allows us to have a more detailed analysis of the millions of chemical compounds. The number of chemical compounds tested is much higher than the potential of the conventional method. This way, we can achieve more accurate results in a larger volume of data. Thus, it is important to discover effective antibiotics in order to further enhance the potential of antibiotics. The artificial intelligence approach provides an opportunity for a revolution in the field of medicine and especially in the discovery of antibiotics, thus allowing the scientific community to control in a short time a large volume of chemicals with significant prognostic accuracy in the effectiveness of antibiotics.

## 4. Conclusions

In recent years, there have been outbreaks of epidemic diseases that unfortunately spread rapidly worldwide and showed unexpectedly high mortality rates. Health-related systems have to face the challenge of increasing their accuracy and sensitivity to ensure that patients are not misdiagnosed. The solution is to use the capabilities of the technology in order to identify potentially suspicious situations and to evaluate patterns that may lead to efficient medical treatment. Thus, recent applications focus on providing accurate epidemic forecasting and utilizing AI algorithms to discover appropriate therapies.

This paper highlights the skills needed to address health problems with the help of AI-related technologies. Thus, the combination of ML, DL, NN, IoT, and CC shows results that are more reliable and at the same time more valid. Obviously, with the mixture of the above technologies, the field of AI can develop various prediction models that will be able to shed light on the possibility of the development of a disease.

As for future work, for predicting the next onset of epidemics, it is necessary to consider a large number of training data, including several parameters. This means not only having the demographic data of a region, a country, or a continent, but also monitoring as much data as possible in each country. Therefore, it is imperative that national or regional surveillance centers be set up first to monitor and analyze locally collected data, such as Centers for Disease Control. In addition, a continental partnership between national centers such as the European Centre for Disease Prevention and Control (ECDPC) is required. Finally, global cooperation between continents could lead to the creation of a global epidemic monitoring center and allow prognostications on viral outbreaks commensurate with the forecasting of other infectious diseases.

**Funding:** This research was funded by the Greek Ministry of Education and Religious Affairs for the project "Enhancing Research and optimizing UOM's administrative operation".

**Institutional Review Board Statement:** Not applicable.

**Informed Consent Statement:** Not applicable.

**Conflicts of Interest:** The authors declare no conflict of interest.

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
