# Peer review of "A Machine Learning-Based Model for Epidemic Forecasting and Faster Drug Discovery"

_applsci, doi:10.3390/app122110766_

Round 1
Reviewer 1 Report
The manuscript is well written. In this article, the authors highlighted the skills needed to address health problems with the help of 368 AI-related technologies. Authors have explored the combination of ML, DL, NN, IoT, and CC, showing good results. However, I would suggest that the author should explore more "Artificial intelligence (AI) in the current scenario. I would further suggest that authors should use an infographic for all the methods mentioned in the methodology section, as it will attract readers.
Author Response
Thank you for your valuable feedback. Indeed, an infographic gives a better experience to readers. Hence, we have added an infographic of the technologies involved in our model and a brief contribution of each to the science of medicine.
Reviewer 2 Report
1. The present paper is confusing as it is submitted to original research and written as a review paper. Authors need to address this first.
2. Methods: where is your unique model to compare with others? This is a major drawback.
3. Did you bring in any new algorithms/machine learning to compare with existing models?
4. Did you compare your model with epidemic forecasting and acceleration in drugs and antibiotics discovery?
5. References: There is scope to include the latest references (2020-2022).
Author Response
According to the first reviewer's feedback we added a code model, Its main advantages are the native support of GPU and ML frameworks, the release of pre-trained models and the functionality to see accuracy and loss in real-time. It supports training and inference through data streams. The Algorithm describes the process of training work that has been run by the ML back-end and shows a sequence diagram of the training process as well as mentioning the advantages. Also added updated references.
Reviewer 3 Report
Current manuscript deals with the analysis of epidemic forecasting and drug discovery based on machine learning-based model. Based on the importance of the topic the manuscript is acceptable after minor revision of the following:
1. Some recent review and research articles on ML for COVID analysis need to be added (10.1016/j.matpr.2020.10.962, 10.1016/j.imu.2021.100564, 10.1186/s12911-021-01742-0, 10.1016/S2589-7500(22)00048-6}.
Author Response
Recent articles from the past two years explaining ML techniques where they can be used to understand and combat this virus have been added as per the reviewer's comments. They also point to effectiveness in diagnosing COVID-19 and predicting mortality risk. Finally, they analyze our ML modeling showing the ability to predict the number of people affected by COVID-19 as a potential threat to humans.
Round 2
Reviewer 2 Report
The authors have made sufficient changes to the original manuscript. I am happy with it.
Author Response
Thank you very much for your helpful comments and the final recommendation.